# A Deeper (Autoregressive) Approach to Non-Convergent Discourse Parsing

**Yoav Tulpan**
Ben Gurion University of the Negev
`yoavtu@post.bgu.ac.il`

**Oren Tsur**
Ben Gurion University of the Negev
`orentsur@bgu.ac.il`

## Abstract

Online social platforms provide a bustling arena for information-sharing and for multi-party discussions. Various frameworks for dialogic discourse parsing were developed and used for the processing of discussions and for predicting the productivity of a dialogue. However, most of these frameworks are not suitable for the analysis of contentious discussions that are commonplace in many online platforms. A novel multi-label scheme for contentious dialog parsing was recently introduced by Zakharov et al. (2021). While the schema is well developed, the computational approach they provide is both naive and inefficient, as a different model (architecture) using a different representation of the input, is trained for each of the 31 tags in the annotation scheme. Moreover, all their models assume full knowledge of label collocations and context, which is unlikely in any realistic setting. In this work, we present a unified model for Non-Convergent Discourse Parsing that does not require any additional input other than the previous dialog utterances. We fine-tuned a RoBERTa backbone, combining embeddings of the utterance, the context and the labels through GRN layers and an asymmetric loss function. Overall, our model achieves results comparable with SOTA, without using label collocation and without training a unique architecture/model for each label. Our proposed architecture makes the labeling feasible at large scale, promoting the development of tools that deepen our understanding of discourse dynamics.

## 1 Introduction

Online discourse has become a major part of modern communication due to the proliferation of online social platforms that allow people to easily share their ideas with a global audience. However, the ease of communication has also led to more heated debates and arguments that sometimes devolve into personal attacks (Arazy et al., 2013; Kumar et al., 2017; Zhang et al., 2018), and increase political and societal polarization (Kubin and von Sikorski, 2021; Lorenz-Spreen et al., 2022).

The ability to parse contentious discussions at a large scale bears practical and theoretical benefits. From a theoretical perspective it would allow the research community at large (social scientists and computational scientists alike) to better track and understand conversational and societal dynamics. From a practical perspective, it was found that early intervention by a human moderator or facilitator can improve the productivity and focus of a discussion (Wise and Chiu, 2011; Chen et al., 2018). Discourse parsing can be the first step in developing assistive moderation tools that can be employed at scale and promote a more productive discourse.

It is commonly argued that the *convergence* of views indicates the success (or productiveness) of a conversation (Barron, 2003; Dillenbourg and Fischer, 2007; Teasley et al., 2008; Lu et al., 2011). This perspective has been reflected in discourse annotation schemes that were proposed through the years (Teasley et al., 2008; Schwarz et al., 2018). However, the equation of *productiveness* with *convergence* is being challenged based on both theoretical and empirical grounds, as non-convergent discussions can be very productive, as they serve as a fruitful venue for the development of dialogic agency (Parker, 2006; Lu et al., 2011; Trausan-Matu et al., 2014; Kolikant and Pollack, 2015; Hennessy et al., 2016; Kolikant and Pollack, 2017).

The non-convergence perspective inspired a novel annotation scheme that was recently introduced by Zakharov et al. (2021). Its organizing principle is *responsiveness*, rather than acceptance and convergence of ideas – a productive discussion is one in which the interlocutors use speech acts that exhibit high responsiveness, while acts of low responsiveness deem the discussion unproductive. It is impor-

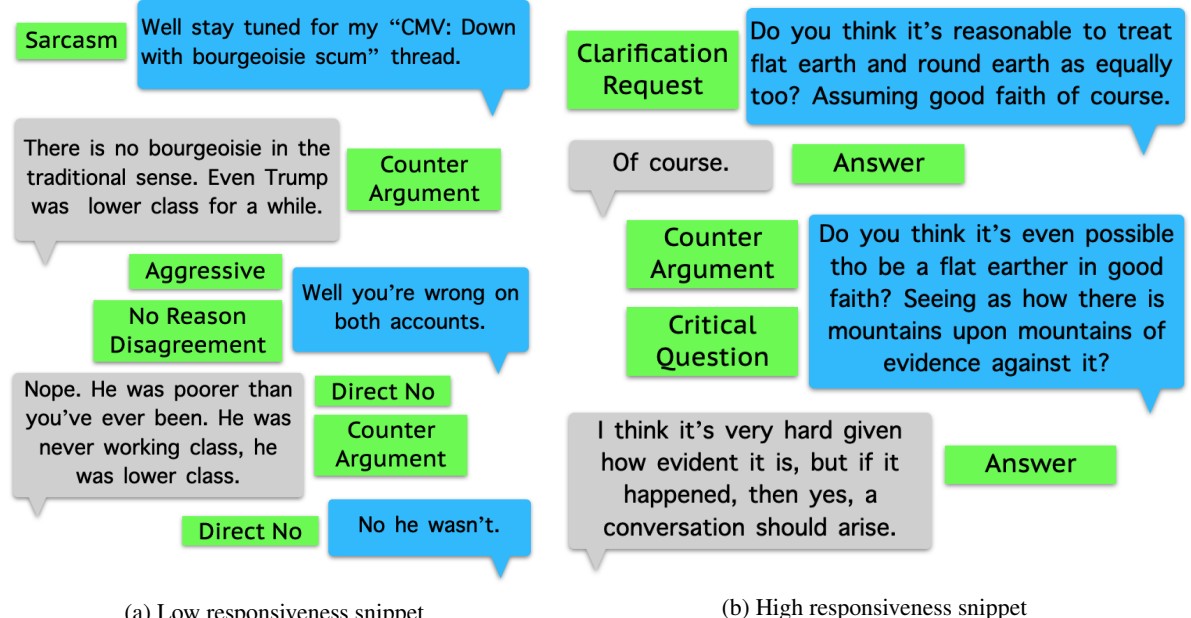

(a) Low responsiveness snippet

(b) High responsiveness snippet

Figure 1: Two annotated snippets extracted from the CMV dataset, displaying low responsiveness (claim: no need for privacy regulation), and high-responsiveness discourse (claim: online cancel culture is ineffective). Labels are indicated in the green rectangles to the left/right of each utterance.

tant to note that responsiveness is not the mere act of producing a response, but the act of responding in good faith. The application of this schema is illustrated by the two snippets in Figure 1. In the short exchange in Figure 1a, the first speaker uses `sarcasm`[1], and later responds `aggressively` to a `Counter Argument`. The dialogue then goes from bad to worse with a series of `Direct No` utterances. The other discussion (Figure 1b) demonstrates how `Counter Argument` and `Critical Question` push for a reasoned answer, even though the topic is highly divisive. Another interesting observation that applies to many online discussions is the way argumentation tends to introduce sub-topics as rhetoric devices[2].

Subscribing to this annotation scheme, the Conversational Discourse Parsing (CDP) task can be viewed as a sequence-of-utterances to sequence-of-sets task: an utterance can be labeled by multiple labels concurrently. For clarity, we provide a brief explanation of the tagset in Section 3. A formal

definition of the computational task is presented in Section 4.1.

The need for a dedicated discourse schema and the development of the tagset were well motivated by Zakharov et al. (2021). The authors released an annotated dataset of $\sim 10K$ utterances and demonstrated the feasibility of learning the annotation task. However, their computational approach suffers from a number of drawbacks: First, they cast the prediction task as a binary classification and trained a model for each tag separately. Second, considering the prediction of tag $l'$ to an utterance $u_i$, they assumed access to an oracle providing complete and accurate knowledge of gold labels of preceding utterances and the correct binary assignment of all other tags for $u_i$. This very strong assumption is not realistic in any real-world scenario. Finally, the results they report were achieved after feature engineering and an extensive grid search on the classifier and the features space. Consequently, each tag is predicted using a different classification framework, based on a uniquely crafted feature set.

In this work, we present *N-CoDiP* – a unified autoregressive transformer for Non-Convergent Discourse Parsing. The model is trained to predict all labels together without using any external knowledge provided by an oracle. N-CoDiP performance (F-score macro and weighted averages) is compara-

---

[1]The sarcastic remark (Figure 1a) follows an argumentation line asserting that privacy is a form of capital (in the Marxist sense) and that it maintains social power imbalance.

[2]The discussion in Figure 1a originated from an opening statement about privacy, discrimination and (social) power imbalance. The discussion in Figure 1b stemmed from a discussion about 'cancel culture' and the reference to 'flat earthers' is a rhetoric device used to establish a common ground.

ble with the best results reported by Zakharov et al. (2021) without suffering from any its drawbacks.

Our proposed model uses the RoBERTa architecture (Liu et al., 2019) as the backbone. We use SimCSE (Gao et al., 2021) for sentence embedding and feed preceding utterances through a Gated Residual Network (GRN) (Lim et al., 2021). The model is fine-tuned using an asymmetric loss function that was recently demonstrated to improve performance in imbalanced multi-label assignment in vision (Ridnik et al., 2021). To the best of our knowledge, this is the first application of this loss function in this domain. We provide a detailed description of the architecture in Section 4. Results and analysis are provided in Section 6.

## 2 Related Work

**Conversational Discourse Parsing** There have been numerous dialog corpora collected and labeled with various schemes to model discourse structure. (Jurafsky et al., 1997) presented the Switchboard-DAMSL dialog act schema on a dataset of cooperative, task-oriented dialogues between pairs of interlocutors in phone conversations (Godfrey et al., 1992). This was extended in (Calhoun et al., 2010) to allow for more thorough analysis of linguistic features in dialog. There have been multiple studies approaching the dialog act classification problem with deep neural networks including transformers, with some emphasizing the importance of integrating context information from previous utterances (Liu et al., 2017; Saha et al., 2019; Santra et al., 2021; Żelasko et al., 2021). The Switchboard-DAMSL corpus is a two party discourse analysis schema, which is different from the multi-party discourse parsing schema presented in (Zakharov et al., 2021) and modeled in this work. Multi-party dialog corpora such as STAC (Asher et al., 2016) as well as the Ubuntu unlabeled corpus (Lowe et al., 2015) and its labeled extension the Molweni discourse relation dataset (Li et al., 2020) are more closely related to the current task, though the discourse is not contentious and the utterances tend to be quite short when compared to messages in the CMV forum debates. Another key difference between these and the CDP corpus is that in the latter, the label scheme is oriented towards a more basic understanding of the components of a productive discourse, while the former is more focused on characterizing basic dialog acts.

**CMV and Discourse** Owing to the high quality of its discussions, CMV discussions are commonly used as a data source for various NLP and social science research, ranging from argument mining to the study of the effects of forum norms and moderation, as well as persuasive text analysis and linguistic style accommodation, e.g., Tan et al. (2016); Khazaei et al. (2017); Musi et al. (2018); Jo et al. (2018); Xiao and Khazaei (2019); Ben-Haim and Tsur (2021); Chandrasekharan et al. (2022).

**Argumentation and argument mining** Argument mining is another related line of research, for a comprehensive survey see (Lawrence and Reed, 2020). Argument mining is done on long-form documents, e.g., Wikipedia pages and scientific papers (Hua and Wang, 2018) or in dialogical contexts, e.g., Twitter, Wikipedia discussion pages, and Reddit-CMV (Tan et al., 2016; Musi et al., 2018; Al Khatib et al., 2018). Argument mining enables a nuanced classification of utterances into discourse acts: socializing, providing evidence, enhancing understanding, act recommendation, question, conclusion, and so forth (Al Khatib et al., 2018). Most of the argument mining work is aimed at identifying stance and opinionated utterance or generating arguments or supportive evidence to end users conducting formal debates (Slonim et al., 2021). Our work is inspired by these works, although our focus is on the way discursive acts reflect and promote responsiveness, rather than simply labeling texts as bearing 'evidence' or posing a 'question'. Moreover, while our focus is contentious non-convergent discussions, we wish to characterize discussions as win-win, rather than a competition.

**Multi-label classification** Regarding imbalanced multi-label classification, the existing approaches include over- and under-sampling the relevant classes, as well as adapting the classification architecture using auxiliary tasks to prevent overfitting to the majority classes (Yang et al., 2020; Tarekegn et al., 2021). Another approach is to apply imbalanced loss functions to neural network models such as weighted cross entropy and focal loss, which is closely related to the Asymmetric loss function incorporated in this work apart from some key improvements detailed in section 4.2.5 (Lin et al., 2017; Ridnik et al., 2021).

## 3 Data

**Change My View (CMV) data**  CMV is self-described as *"A place to post an opinion you accept may be flawed, in an effort to understand other perspectives on the issue. Enter with a mindset for conversation, not debate."*[3]  Each discussion thread in CMV evolves around the topic presented in the submission by the Original Poster (OP). Each discussion takes the form of a conversation tree in which nodes are utterances. A directed edge $v \leftarrow u$ denotes that utterance $u$ is a direct reply to utterance $v$. A full branch from the root to a leaf node is a sequence of utterances which reflects a (possibly multi-participant) discussion. CMV is heavily moderated to maintain a high level of discussion. CMV data has been used in previous research on persuasion and argumentation, see a brief survey in Section 2.

**Annotation scheme tagset**  The Contentious Discourse Parsing tag schema developed by Zakharov et al. (2021) consists of 31 labels that fall under four main categories: discursive acts that promote further discussion; discursive acts exhibiting or expected to cause low responsiveness; tone and style; explicit disagreement strategies. For convenience, the full schema and the labels' definitions are provided in Appendix B.

The annotation scheme allows a collocation of labels assigned to the same utterance as some labels reflect style while others reflect the argumentative move. For example, the utterance *"well you're wrong on both accounts."* (Figure 1a) carries an `Aggressive` tone, providing `No Reason` for the disagreement it conveys.

**The annotated dataset**  The dataset released[4] by (Zakharov et al., 2021) is composed of 101 discussion threads from CMV. These threads (discussion trees) have a total of 1,946 branches composed of 10,599 utterances (nodes) made by 1,610 unique users. The number of labels assigned to the nodes in the dataset is 17,964.

---

[3] https://www.reddit.com/r/changemyview/wiki/index (accessed 1/17/23)
[4] The dataset was made available here: http://bit.ly/3XIMeD1 (last accessed: 1/17/23).

## 4 Computational Approach

### 4.1 Task Definition

We define the discourse parsing classification problem as follows: Given a tagset $T$ and a sequence of utterances $U = u_1, ..., u_n$: find a corresponding sequence of labels $L = l_1, ..., l_n$ such that it maximizes the probability $P(L|U)$. It is important to note that each $l_i$ is actually a *set* of labels from the $T$ such that $l_i \subset T$, making this a sequence to sequence-of-sets task. The sequence of utterances is processed sequentially in an autoregressive manner. That is, when tagging $u_i$ the model already processed $u_1$ through $u_{i-1}$ and $u_{j>i}$ are masked.

### 4.2 N-CoDiP Architecture and Components

Given a sequence of utterances $u_1, ..., u_n$, utterance $u_i$ is processed along with its context $c_i$ – the utterances preceding it ($u_1, ..., u_{i-1}$). First, we use the pretrained model to get two embedding vectors $\vec{u_i}$ and $\vec{c_i}$ representing $u_i$ and $c_i$, respectively. We then use two GRN blocks: The first combines $\vec{c_i}$ with $\vec{l^{i-1}}$, the label embeddings vector produced in the previous iteration (processing $u_{i-1}$). The second GRN block combines the resulting vector with $\vec{u_i}$ for a combined representation. This representation is passed to a block of MLP classifiers which produce $\hat{l_i}$, a vector assigning the likelihood of each tag $t \in T$ for $u_i$. An illustrative figure of the model is provided in Figure 2. In the remainder of the section we present the components of the N-CoDiP architecture in detail.

#### 4.2.1 Text Representation

The representation of the target utterance $u_i$ and the context utterances $c_i$ are produced separately in a slightly different ways. $\vec{u_i}$, the representation of $u_i$ is simply the $[CLS]$ token vector obtained by passing $u_i$ to the pretrained model. The context representation $\vec{c_i}$ is the $[CLS]$ of the concatenated word-tokens of the context utterances, using the $[SEP]$ token to separate between utterances in order to allow context utterances to attend to each other. That is, the context utterances are passed as a sequence $u_{i-k}[SEP]u_{i-k+1}[SEP]...[SEP]u_{i-1}$, where $k$ is the length of the context and $u_j$ is the sequence of tokens in the $j^{th}$ utterance.

#### 4.2.2 Context's Label Embedding

We define a label embedding function $Emb(\cdot) \in \mathbb{R}^d$ where $d$ is the transformer embedding dimension (in our case, 768). In cases where a previous

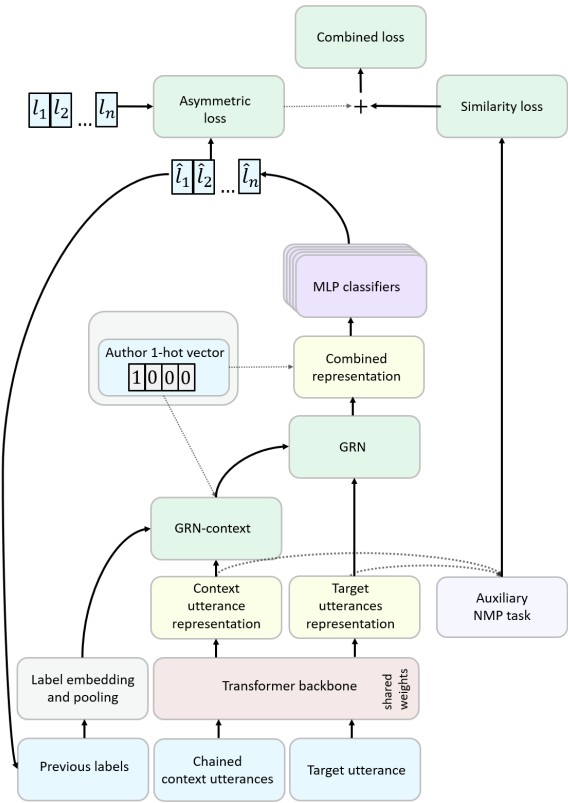

Figure 2: N-CoDiP architecture. Dotted arrows indicate optional components.

utterance is unlabeled, we add an additional embedding that represents an *untagged* context utterance. We combine the label embeddings of the multiple utterances in the context using mean-pooling.

### 4.2.3 Context Integration with GRNs

Gated Residual Networks (GRN) (Lim et al., 2021) were recently proposed in order to combine a primary input vector with context vectors of multiple types and unknown relevance. GRNs were demonstrated to be especially beneficial when the dataset is relatively small and noisy.

Formally, given a vector $x$ and a context vector $c$:

$$GRN(x, c) =$$
$$LayerNorm(x + GatedLinear(\eta_1))$$
$$\eta_1 = W_1\eta_2 + b_1$$
$$\eta_2 = ELU(W_2 x + W_3 c + b_2)$$
$$GatedLinear(\gamma) =$$
$$\sigma(W_4\gamma + b_4) \odot (W_5\gamma + b_5)$$

Where $W_i(\cdot) + b_i$ is a linear transformation maintaining the input dimension $d$, and $ELU(\cdot)$ is an Exponential Linear Unit (Clevert et al., 2015).

We use GRNs to combine the textual embedding of the context ($\vec{c_i}$) with pooled label embeddings ($\vec{l_i}$), and again to combine the result with $\vec{u_i}$, the embedding vector of the target utterance.

### 4.2.4 Multi-head MLP

In the final layer, the combined representation is passed to $d$ independent MLP heads, with $d$ being the number of labels in the tagset. Given the last hidden layer output $z$, the model's prediction for the $i$'th label is:

$$\hat{l}_i = \sigma(W_{i,2}ReLU(W_{i,1}z + b_{i,1}) + b_{i,2})$$

### 4.2.5 Asymmetric Loss

The Asymmetric Loss was recently developed to handle unbalanced multi-label classification tasks in the field of computer vision (Ridnik et al., 2021). The asymmetric loss applies a scaling decay factor to the loss in order to focus on harder examples. However, different decay factors are used for instances with positive and negative gold labels: a larger decay factor ($\gamma_- > \gamma_+$) to the negative examples. Also, it employs a hard lower cutoff $m$ for model confidence scores to discard too-easy examples.

Asymmetric loss was used for relation extraction between entities in a given document by (Li et al., 2021), but is still underexplored in the NLP context and was never used for conversational discourse parsing.

It allows the model to learn the task despite positive to negative label imbalances, which are often a hindrance to neural network performance. The $AL$ (Asymmetric Loss) function is defined over the positive cases $L_+$ and the negative cases $L_-$:

$$AL(\hat{l}_i, l_i) = \begin{cases} (1 - \hat{l}_i)^{\gamma_+}log(\hat{l}_i) & l_i \in L_+ \\ l_m^{\gamma_-}log(1 - l_m) & l_i \in L_- \end{cases}$$

$l_m = max(\hat{l}_i - m, 0)$, and $m$ is the lower hard cutoff of model confidence scores for negative labels.

### 4.2.6 Auxiliary Next Message Prediction Task

Incorporating an auxiliary prediction task to the training pipeline often improves results, especially over relatively small datasets for which pretrained models tend to overfit (Chronopoulou et al., 2019; Schick and Schütze, 2021). Drawing inspiration from (Henderson et al., 2020), we incorporate Next Message Prediction (NMP) as an auxiliary task. In

the NMP the model maximizes the cosine similarity of two consecutive messages in the conversation tree, and minimizes that of non-consecutive ones. That is, the training objective of this auxiliary task is to minimize $L_{NMP}$, defined as:

$$L_{NMP} = \sum_{i=1}^{k} \sum_{j=1}^{k'} S(u_i, u_j) - \sum_{i=1}^{k} S(u_i, u_j)$$

Where $S$ is a similarity function (we use cosine similarity), $k$ is the batch size for the main Discourse Parsing (DP) task, and $k'$ is the number of negative samples, which are simply the other utterances in the batch. We also attempted to add more challenging negative samples, i.e., samples that are sampled from the same conversation tree as $u_i$ and are therefore assumed to belong to the same semantic domain. The final loss function to be minimized in training is:

$$L = \alpha L_{DP} + (1 - \alpha) L_{NMP}$$

$L_{DP}$ is the Asymmetric loss described in section 4.2.5, and $\alpha \in [0.95, 0.99]$ is a weighting factor for the different objectives.

### 4.2.7 Speakers' Turn Taking

We expect that the conversational dynamics in a dialogue of only two speakers are may be different than those in a multi-speaker dialogue. Moreover, even in a multi-speaker dialogue, the discourse between speakers $A$ and $B$ may be different that the discourse between $A$ and $C$. We therefore add $k+1$ one-hot vectors representing the the speakers of the target utterance $u_i$ and the $k$ preceding utterances used for context. That is, given $k = 3$ and the sequence of utterances $u_{i-3}^A u_{i-2}^B u_{i-1}^C u_i^A$ (postscript denotes the speaker), we get the following vectors:

$$[1, 0, 0, 0], [0, 1, 0, 0], [0, 0, 1, 0], [1, 0, 0, 0]$$

indicating that $u_i$ and $u_{i-3}$ were produced by the same speaker (A), while $u_{i-2}$ and $u_{i-1}$ where produced by two other speakers (B and C). These vectors were concatenated and appended to the final combined representation vector.

## 5 Experimental Settings

**Baselines**   We compare our N-CoDiP architecture to previously reported results in (Zakharov et al., 2021). We focus on two sets of reported results:

1. **Exhaustive Grid (X-Grid)** The best results reported by Zakharov et al. (2021) achieved using a different model for each label, extensive feature engineering, external resources (LIWC, DPTB discourse labels), an Oracle providing preceding and collocated labels and exhaustive grid-search in a binary classification setting (per label).

2. **Zakharov Transformer (Z-TF)** The same Transformer architecture used by (Zakharov et al., 2021) applied in a "clean" setting, that is, without the use of an oracle or special (external) features. The use of this baseline allows a proper evaluation of our model against prior work.

**Pretrained Models**   We consider two pretrained models for text representation: the vanilla RoBERTa (Liu et al., 2019) and the RoBERTa-SimCSE that was optimized for sentence embedding (Gao et al., 2021). We indicate the pretrained model that is used in subscript: $CoDiP_V$ for the Vanilla RoBETRa and $CoDiP_{CSE}$ for the SimCSE version.

**Evaluation Metrics**   Keeping in line with previous work we use F-score ($F_1$) for individual labels. We report both macro and weighted F-scores results aggregated by label category. *Macro* F-score being the mean score, and *weighted* being the mean weighted according to the support of each class:

$$F_{Macro}(F_1, ..., F_k) = \frac{\sum_{i=1}^{k} F_i}{k}$$

$$F_{Weighted}(F_1, ..., F_k) = \sum_{i=1}^{k} F_i \cdot w_i$$

where $k$ is the number of labels in a particular label category (e.g., Promoting Discourse, Disagreement Strategies). $w_i$ is the prior probability of a specific label $l_i$ being true in the dataset, which is comprised of $n$ samples:

$$w_i = \frac{\sum_{i=1}^{n} \mathbb{1}_{l_i=1}}{n}$$

The prior probabilities are presented in table 3 in Appendix A.

**Execution Settings**   We trained the N-CoDiP model for 4 epochs optimized using the AdamW optimizer (Loshchilov and Hutter, 2018) with a

| Category | N-CoDiP$_{CSE}^{AL}$ | | N-CoDiP$_{CSE}^{BCE}$ | | N-CoDiP$_{V}^{AL}$ | | Z-TF | | X-Grid | |
|---|---|---|---|---|---|---|---|---|---|---|
| All | **0.397**[†] | **0.573** | 0.371 | 0.563 | 0.378 | 0.565 | 0.113 | 0.338 | 0.382 | 0.606[†] |
| Promoting Discussion | **0.461** | **0.709** | 0.426 | 0.692 | 0.439 | 0.690 | 0.158 | 0.546 | 0.560[†] | 0.833[†] |
| Low Responsiveness | **0.312**[†] | **0.337**[†] | 0.276 | 0.304 | 0.284 | 0.309 | 0.058 | 0.057 | 0.308 | 0.335 |
| Tone and Style | **0.346**[†] | **0.370**[†] | 0.320 | 0.352 | 0.334 | 0.361 | 0.054 | 0.064 | 0.304 | 0.326 |
| Disagreement Strategies | **0.422**[†] | **0.507**[†] | 0.408 | 0.497 | 0.407 | 0.499 | 0.142 | 0.170 | 0.370 | 0.451 |

Table 1: Average F-scores per label category for each model. Values are arranged as (*Macro*, *Weighted*) pairs. N-CoDiP architectures differ in the loss function used: Asymmetric Loss ($AL$) or Binary Cross Entropy ($BCE$), and the pretrained model used: Contrastive Sentence Embedding ($CSE$) or the vanilla RoBERTa ($V$); Z-TF is the BERT architecture used by Zakharov et al. (2021); X-Grid are the best results reported in prior work using an oracle and applying an exhaustive grid search over parameters and models for each of the labels. A † indicates best results overall. Best results achieved by a transformer architecture without an oracle or feature engineering are in bold face.

batch size of 32. We used a linear warm up and decay on the learning rate, with the warm up period consisting of first $30\%$ of the training iterations reaching maximal $\eta = 10^{-5}$ learning rate and decaying back to zero over the remaining $70\%$ of the training iterations. We restrict our experimentation to contexts of up to $k$ utterances and set $k = 4$. For the Asymmetric loss we used the default parameters $\gamma_+ = 1; \gamma_- = 4; m = 0.05$.

**Computational Cost** We trained our final implementation of the model 20 times (4 model variations × 5 fold cross validation), as well as additional implementations during its development, each taking between 2 and 3 hours on a Nvidia GeForce 12GB GPU. The model contains 130,601,503 parameters.

# 6 Results and Analysis

## 6.1 Results

All reported results are the average of a 5-fold cross validation. Partition of the data in each fold was done based on discussion trees rather than conversation branches in order to avoid leakage from the train set to the test set.

Macro and weighted F-Score over the whole tagset and by label's categories are provided in Table 1. Prior probabilities and detailed results for each label are omitted for clarity and due to space constraints but are available at Appendix A.

The results that were reported by prior work (X-Grid) are presented as a guide, but are shaded since the X-Grid setting does not allow a fair comparison. We expand on this in the discussion.

N-CoDiP$_{CSE}^{AL}$ consistently outperforms all other unified models trained to predict all labels without any prior or external knowledge in both Macro and weighted scores. Moreover, N-CoDiP$_{CSE}^{AL}$ outperforms X-Grid over three out of the four label categories (Low Responsiveness, Tone & Style, Disagreement Strategies), and obtains a higher Macro average F-score aggregated over all labels.

Evaluating the impact of the loss function (Asymmetric vs. Binary Cross-entropy) we find that the asymmetric loss is consistently better. We also find that the most significant improvements are achieved over the Low Responsiveness, Tone & Style categories, for which the priors are relatively low (see Table 3 in Appendix A). This is also evident by comparing the gains in the macro averages vs the gains in the weighted average: 0.026 and 0.01, respectively.

Also, for most labels the use of the pretrained RoBERTa-SimCSE achieves better results than the vanilla RoBERTa, gaining 0.019 macro-F points, and only 0.012 points in the weighted score.

| Category | N-CoDiP$_{k=1}$ | | N-CoDiP$_{k=4}$ | |
|---|---|---|---|---|
| All | **0.397** | **0.573** | 0.389 | **0.573** |
| Promoting Disc. | **0.461** | **0.709** | 0.426 | 0.699 |
| Low Resp. | **0.312** | **0.337** | 0.298 | 0.328 |
| Tone and Style | **0.346** | **0.370** | 0.338 | 0.328 |
| Disagreement Str. | **0.422** | **0.507** | **0.422** | 0.506 |

Table 2: Average F-scores per label category for the N-CoDiP model given $k = 1$ context length and $k = 4$ context length. Values are (*Macro*, *Weighted*) pairs.

## 6.2 Discussion

**N-CoDiP vs. X-grid** While N-CoDiP achieves best results in most cases, the X-Grid achieves a higher weighted score on the aggregation of all labels, and significantly outperforms CoDiP in the Promoting Discussion category. It is important to reiterate that the X-Grid setting does not allow a fair comparison. Not only were each of

the X-Grid results obtained by a different classifier based on different feature set, it combines heavy feature engineering of external reasources such as LIWC categories (Tausczik and Pennebaker, 2010), DPTB labels (Prasad et al., 2008; Nie et al., 2019), an Oracle providing preceding and collocated labels (classification is binary per label), and an exhaustive grid search over the model family, features, and hyper parameters. In contrast, the rest of the results in Table 1 are achieved using a single unified model without incorporating any auxiliary resources except RoBERTa, and no Oracle hints.

**N-CoDiP vs. Z-TF** Although the results presented above establish the effectiveness of a single unified model, we observe a stark difference in performance between all variants of the N-CoDiP architecture and the Z-TF. This difference begs the question what in the architecture makes such an impact, given both approaches rely on the same pretrained BERT based architecture. We hypothesize that the combination of the multi-head classifier and the Asymmetric loss objective (Sections 4.2.4 and 4.2.5) drive CoDiP performance up. The individual classifiers add another layer which enables the model to learn a unique final hidden representation for each label. We have found this to be quite effective in mitigating the label bias. Indeed, we observe that even though Z-TF is inferior to CoDiP, it does perform reasonably well on the most frequent label (`CounterArgument`; $p = 0.635$, see Table 3 in Appendix A). In addition, the asymmetric loss function provides significant gains for less common labels, promoting the hypothesis that the poor Z-TF performance stems from a label imbalance, a common issue in multi-class neural network based classifiers (Xiao et al., 2019).

Finally, unlike the autoregressive architecture of the CoDiP models, Z-TF naively uses the Transfomer as a non-autoregressive classifier. Consequently, while it processes preceding utterances to provide context to the target utterance, it does not leverage the labels that were predicted for the context.

**Context length and multi-modality** Surprisingly, we found that adding as many context utterances as the encoder can take resulted in *degraded* performance, comparing to using only the single immediate context ($k = 1$). A comparison between context length of 1 and 4 is provided in Table 2. Similarly, we find it surprising that adding the author turn-taking information (see Section 4.2.7) did

not yield any improvement. We believe that the ways contexts (and different contextual signals) are integrated and attended to should be further investigated in order to leverage the full potential of the information encoded in the context.

**The unimpressive contribution of auxiliary task** Incorporating an auxiliary prediction task to the training pipeline is reported to often improve results, especially when fine-tuning over relatively small datasets (Chronopoulou et al., 2019; Henderson et al., 2020; Schick and Schütze, 2021). We experimented with a number of settings for utterance proximity prediction to no avail – results were not improved in any significant way. We plan to explore this further in the future.

## 7 Conclusion and Future Work

Theoretical framework and empirical evidence motivates the need for a discourse annotation schema that reflects discursive moves in contentious discussions. We introduced N-CoDiP, a unified Non-Convergent-Discussion Parser that outperforms previous work in a discourse parsing task based on the scheme that was recently developed and shared by Zakharov et al. (2021).

We have demonstrated that using GRN layers, previously used for multi-horizon time-series forecasting by Lim et al. (2021) and an asymmetric loss function, previously used in computer vision by Ridnik et al. (2021) is especially beneficial to the task at hand, given the relatively small dataset, the imbalanced tagset, and the multi-label setting.

Future work will take theoretical and computational trajectories. A robust error analysis will be done with respect to the theoretical framework behind the annotation scheme. Computationally, we will investigate better ways to better leverage the abundance of structured unlabeled data (thousands of discussion on CMV and other platforms) as an auxiliary task, and achieve a better integration of the context turn-taking structure with the model.

## 8 Limitations

The main limitation of the paper is the size of the dataset, given the large and imbalanced tagset and the complex and nuanced discourse annotation scheme. We believe that expanding the dataset and maybe reconsidering some nuances in the annotation scheme would mitigate the issue.

## 9 Ethics and Broader Impact

This paper is submitted in the wake of a tragic terrorist attack perpetrated by Hamas, which has left our nation profoundly devastated. On October 7, 2023, thousands of Palestinian terrorists infiltrated the Israeli border, launching a brutal assault on 22 Israeli villages. They methodically moved from home to home murdering more than a thousand innocent lives, spanning from infants to the elderly. In addition to this horrifying loss of life, hundreds of civilians were abducted and taken to Gaza. The families of these abductees have been left in agonizing uncertainty, as no information, not even the status of their loved ones, has been disclosed by Hamas.

The heinous acts committed during this attack, which include acts such as shootings, sexual assaults, burnings, and beheadings, are beyond any justification.

We fervently call for the immediate release of all those who have been taken hostage and urge the academic community to unite in condemnation of these unspeakable atrocities committed in the name of the Palestinian people. We call all to join us in advocating for the prompt and safe return of the abductees, as we stand together in the pursuit of justice and peace.

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

## A   F-scores by Label

| Label/Category | N-CoDiP | N-CoDiP$_{BCE}$ | N-CoDiP$_{BASE}$ | Z-TF | X-Grid | Priors |
|---|---|---|---|---|---|---|
| **1. Promotes discussion** | | | | | | |
| ViableTransformation | **0.118** | 0.09 | 0.092 | 0 | 0.158$^\dagger$ | 0.01 |
| Answer | 0.413 | 0.366 | 0.397 | **0.522**$^\dagger$ | 0.522$^\dagger$ | 0.014 |
| Extension | **0.286** | 0.258 | 0.263 | 0.507 | 0.549$^\dagger$ | 0.022 |
| AttackValidity | **0.506** | 0.435 | 0.48 | 0.143 | 0.51$^\dagger$ | 0.028 |
| Moderation | **0.353** | 0.277 | 0.326 | 0.027 | 0.42$^\dagger$ | 0.036 |
| RequestClarification | **0.488** | 0.482 | 0.471 | 0.160 | 0.731$^\dagger$ | 0.038 |
| Personal | 0.646 | 0.644 | **0.654**$^\dagger$ | 0.066 | 0.396 | 0.046 |
| Clarification | **0.524** | 0.466 | 0.459 | 0 | 0.817$^\dagger$ | 0.109 |
| CounterArgument | **0.818** | 0.813 | 0.805 | 0.775 | 0.939$^\dagger$ | 0.635 |
| **2. Low responsiveness** | | | | | | |
| NoReasonDisagreement | **0.349** | 0.284 | 0.266 | 0 | 0.4$^\dagger$ | 0.01 |
| AgreeToDisagree | **0.39**$^\dagger$ | 0.261 | 0.3 | 0 | 0.2 | 0.014 |
| Repetition | 0.118 | 0.118 | **0.136** | 0 | 0.161$^\dagger$ | 0.016 |
| BAD | 0.217 | 0.256 | **0.257**$^\dagger$ | 0 | 0.114 | 0.018 |
| NegTransformation | **0.169** | 0.131 | 0.151 | 0 | 0.406$^\dagger$ | 0.024 |
| Convergence | **0.630**$^\dagger$ | 0.606 | 0.593 | 0.108 | 0.565 | 0.028 |
| **3. Tone and Style** | | | | | | |
| WQualifiers | **0.351**$^\dagger$ | 0.274 | 0.343 | 0.029 | 0.118 | 0.024 |
| Ridicule | **0.236**$^\dagger$ | 0.193 | 0.207 | 0.029 | 0.11 | 0.029 |
| Sarcasm | 0.212 | **0.216**$^\dagger$ | 0.209 | 0 | 0.164 | 0.048 |
| Aggressive | **0.27**$^\dagger$ | 0.251 | 0.265 | 0 | 0.17 | 0.051 |
| Positive | 0.532 | **0.541**$^\dagger$ | 0.515 | 0.19 | 0.336 | 0.058 |
| Complaint | **0.475**$^\dagger$ | 0.449 | 0.467 | 0.077 | 0.343 | 0.064 |
| **4. Disagreement Strategies** | | | | | | |
| Alternative | **0.192**$^\dagger$ | 0.178 | 0.184 | 0 | 0.133 | 0.018 |
| RephraseAttack | 0.179 | 0.132 | **0.183**$^\dagger$ | 0 | 0.077 | 0.022 |
| DoubleVoicing | 0.162 | 0.146 | **0.179**$^\dagger$ | 0 | 0.179$^\dagger$ | 0.026 |
| Softening | **0.293** | 0.265 | 0.288 | 0.014 | 0.379$^\dagger$ | 0.029 |
| Sources | **0.779** | 0.774 | 0.746 | 0.730 | 0.884$^\dagger$ | 0.045 |
| AgreeBut | 0.473 | **0.481**$^\dagger$ | 0.459 | 0 | 0.106 | 0.058 |
| Irrelevance | **0.286**$^\dagger$ | 0.262 | 0.22 | 0 | 0.172 | 0.059 |
| Nitpicking | 0.760 | 0.763 | **0.786** | 0.447 | 0.79$^\dagger$ | 0.061 |
| DirectNo | **0.458**$^\dagger$ | 0.443 | 0.412 | 0 | 0.259 | 0.08 |
| CriticalQuestion | **0.636** | 0.635 | 0.618 | 0.224 | 0.722$^\dagger$ | 0.128 |

Table 3: Mean 5-fold cross validation F-scores for the individual labels in the tag-set. N-CoDiP architectures differ in the loss function used: Asymmetric Loss (AL) or Binary Cross Entropy (BCE), and the pretrained model used: Contrastive Sentence Embedding (CSE) or the vanilla RoBERTa (V ); Z-TF is the BERT architecture used by Zakharov et al. (2021); X-Grid are the best results reported in prior work using an oracle and applying an exhaustive grid search over parameters and models for each of the labels. A † indicates best results overall. Best results achieved by a transformer architecture without an oracle or feature engineering are in bold face. Prior probabilities included.

# B   Complete Tagset and Label definitions

| Description | Tag |
|---|---|
| **1. Discursive moves that potentially promote the discussion** | |
| Moderating/regulating, e.g. "let's get back to the topic" | Moderation |
| Request for clarification | RequestClarification |
| Attack on the validity of the argument ("Who says?") | AttackValidity |
| Clarification of previous statement (utterance) | Clarification |
| Informative answer of a question asked (rather than clarifying ) | Answer |
| A disagreement which is reasoned, a refutation. Can be accompanied by disagreement strategies | CounterArgument |
| Building/extending previous argument. The speaker takes the idea of the previous speaker and extends it. | Extension |
| A viable transformation of the discussion topic | ViableTransformation |
| Personal statement "this happened to me") | Personal |
| **2. Moves with low responsiveness** | |
| Severe low responsiveness: continuous squabbling | BAD |
| Repeating previous argument without any real variation | Repetition |
| Response to ancillary topic / derailing the discussion | NegTransformation |
| Negation/disagreement without reasoning | NoReasonDisagreement |
| Convergence towards previous speaker | Convergence Agreement |
| The issue is deemed unsolvable by the speaker | AgreeToDisagree |
| **3. Tone and style** | |
| **3.1 Negative tone and style** | |
| Aggressive and Blatant "this is stupid" | Aggressive |
| Ridiculing the partner (or her argument) | Ridicule |
| Complaining about a negative approach "you were rude to me" | Complaint |
| Sarcasm/ cynicism /patronizing | Sarcasm |
| **3.2 Positive tone and style** | |
| Attempts to reduce tension: respectful, flattering, etc. | Positive |
| Weakening qualifiers e.g. "I'm not an expert in this topic..." | WQualifiers |
| **4. Disagreement strategies** | |
| **4.1 Easing tension** | |
| Softening the blow of a disagreement. | Softening |
| Partial disagreement "I disagree only with one part of your text" | AgreeBut |
| Explicitly taking into account other participants' voices | DoubleVoicing |
| Using an external source to support a claim | Sources |
| **4.2 Intensifying tension** | |
| Reframing or paraphrasing the previous comment | RephraseAttack |
| Critical question, phrasing the (counter) argument as a question | CriticalQuestion |
| Offering an alternative without direct refutation | Alternative |
| Direct disagreement ("I disagree", "this is simply not true") | DirectNo |
| Refutation focuses on the relevance of previous claim | Irrelevance |
| Breaking previous argument to pieces without real coherence | Nitpicking |

Table 4: Copied from Zakharov et al. (2021).