# OpenReview forum: "A Deeper (Autoregressive) Approach to Non-Convergent Discourse Parsing"
_EMNLP/2023/Conference — EMNLP 2023 Main_

### Official Review · Reviewer_brxV · 2023-08-03

**Soundness:** 4

**Excitement:**

3: Ambivalent: It has merits (e.g., it reports state-of-the-art results, the idea is nice), but there are key weaknesses (e.g., it describes incremental work), and it can significantly benefit from another round of revision. However, I won't object to accepting it if my co-reviewers champion it.

**Paper Topic And Main Contributions:**

In this paper, the authors proposes a unified model to parse specific kind of discourse (non-convergent ones), based on a fine-tuning of ROBERTa.
They achieve State of the art results, with limited informations.


**Reasons To Accept:**

- The task is very complicated and there are not so much data for it.
- the other conduct correctly their experiment and they have fair enough limitation remarks.


**Reasons To Reject:**

- yet another experiment which slightly improve sota
- Even if the limitation are stated, they are extremely important: the lack of data lead to a non reusable model.
- annotation system w-should be more discussed and adapted.


**Reproducibility:**

3: Could reproduce the results with some difficulty. The settings of parameters are underspecified or subjectively determined; the training/evaluation data are not widely available.

**Reviewer Confidence:**

4: Quite sure. I tried to check the important points carefully. It's unlikely, though conceivable, that I missed something that should affect my ratings.

---

> ### Author Rebuttal · Authors · 2023-08-28
>
> *General*: We thank the reviewers for their time and comments and we are happy that all reviewers agree about the soundness, reproducibility and other merits of the work.
>
> Detailed response:
>
> *Improving SOTA*: The model we propose should not be compared to SOTA only in terms of the numbers. Rather, it is a completely different approach that is not only more efficient and elegant - it does not require an ORACLE “leaking” information (gold labels of context) to the classifier.
>
> *Data*: In the paper we have demonstrated the applicability of the model to this complex task, advocating for the annotation of more data (under process), the promise in this line of research, and the benefits for scholars in other disciplines.
>
> *Annotation protocol*: We used the annotation scheme and data that were developed and provided by Zakharov et al., 2021.The tagset and the rationale behind it, the annotation protocol, and the dataset are presented  at length in Zakhariv et al. and in the accompanying repository (URL in footnote 4 in their paper). In our submission we only provide the necessary background. The reader is referred to the original paper and report by Zaharov for the finer details.

---

### Official Review · Reviewer_MyNg · 2023-08-11

**Soundness:** 3

**Excitement:**

3: Ambivalent: It has merits (e.g., it reports state-of-the-art results, the idea is nice), but there are key weaknesses (e.g., it describes incremental work), and it can significantly benefit from another round of revision. However, I won't object to accepting it if my co-reviewers champion it.

**Paper Topic And Main Contributions:**

The author(s) proposed a new model N-CoDiP for Non-Convergent Discourse Parsing for contentious dialogues, which achieved comparable results with the SOTA on Change My View (CMV) dataset. It also offers the following benefits compared to SOTA: 1) no feature engineering 2) a unified model instead of separate models for different labels 3) no requirement of gold labels of previous utterances.

The task was framed as an utterance-level multi-label classification task, where each utterance will be assigned one or more labels (e.g., Aggresive, No Reason Disagrrment). The dataset - Change My View (CMV) data, is taken from a previous paper. It contains multi-party dialogues constructed based on "ChangeMyView" subreddit with utterance-level annotations.

The proposed model N-CoDiP utilizes pre-trained transformers to get the embeddings of the current utterance and previous context, then aggregate the representations along with the embeddings of the label of the previous utterance using Gated Residual Networks (GRN). Asymmetric Loss was adopted to handle the imbalance of class labels.

The authors also explored the effectiveness of one-hot speaker representation, pretrained model enhanced with Contrastive Sentence Embedding and the auxiliary next message prediction task.

**Questions For The Authors:**

Question A: How do you decide the proper value of \alpha at the end of 4.2.6?
Question B: What is the standard deviation of 5-fold cross validation?
Note: these two questions are well explained by the authors in the rebuttal stage.

**Reasons To Accept:**

The proposed model N-CoDip provides a practical solution for discourse parsing for non-convergent multi-party dialogues. It achieves comparable results with SOTA but requires less resources and is simpler.

**Reasons To Reject:**

Baseline model Z-TF is too simple. After reading the original paper (Zakharov et al., 2021) that utilises Z-TF (lack of details of the Z-TF model in this paper), my understanding is that it only uses the current utterance for classification. It will be better to try more advanced baselines that leverage both the context (previous utterances) and the current utterance. This context-aware baseline is essential to prove that your model componets are necessary, including: Gated Residual Networks (GRN), Label embedding, Auxiliary NMP task.

**Reproducibility:**

3: Could reproduce the results with some difficulty. The settings of parameters are underspecified or subjectively determined; the training/evaluation data are not widely available.

**Reviewer Confidence:**

4: Quite sure. I tried to check the important points carefully. It's unlikely, though conceivable, that I missed something that should affect my ratings.

**Typos Grammar Style And Presentation Improvements:**

It would be better to make the labels smaller and use a different style compared to utterances in Figure 1 to make the following aspects clear: 1) the discrimination between utterances and labels; 2) the correspondence between utterances and labels

---

> ### Author Rebuttal · Authors · 2023-08-28
>
> *General*: We thank the reviewers for their time and comments and we are happy that all reviewers agree about the soundness, reproducibility and other merits of the work.
>
> Detailed response:
> *General*: We thank the reviewers for their time and comments and we are happy that all reviewers agree about the soundness, reproducibility and other merits of the work.
>
> Detailed response:
>
> *Z-TF Baseline*: The Z-TF is naive in the sense that it ignores previous utterances, yet it establishes the baseline obtained by a strong text classifier (BERT). We actually implemented versions that naively accept previous utterances (with separators) when input length permits, or previous gold labels, these results are excluded due to space constraints and our choice to highlight other empirical aspects and results. The GRN ablation results were omitted for similar reasons.
> We do provide results for various context length (Table 2) and discuss it and the contribution of the auxiliary task in Section 6.2. We report on the  gains obtained by the asymmetric-loss in Table 1.
>
>
> *Alpha values*: We naively experimented with some values and noticed that smaller alpha values resulted in fluctuations and in catastrophic forgetting.
>
> *X-val STD*: For convenience, we will reply to the question with the averages of the standard deviations: The N-CoDiP model's per-category 5-fold results (Table 1, first column) have a mean standard deviation of 0.169 with a nearly identical median (0.168). The per-label 5-fold results (Appendix A) have a mean standard deviation of  0.047 with a median of 0.044.
> In both cases, there are no significant outliers in the 5-fold results.

---

### Official Review · Reviewer_KWYb · 2023-08-12

**Typos Grammar Style And Presentation Improvements:** In section 4.2.7, line 6, the term "1…
**Soundness:** 3

**Excitement:**

3: Ambivalent: It has merits (e.g., it reports state-of-the-art results, the idea is nice), but there are key weaknesses (e.g., it describes incremental work), and it can significantly benefit from another round of revision. However, I won't object to accepting it if my co-reviewers champion it.

**Paper Topic And Main Contributions:**

This paper focuses on the analysis of contentious discussions that are commonplace in many online platforms. A unified model for Non-Convergent Discourse Parsing that does not require any additional input other than the previous dialog utterances is proposed. The paper claims that the model achieves results comparable with SOTA.

**Reasons To Accept:**

1) The paper is well-written and easy to read.

2) The experimental results of the proposed methods including several variants are comprehensive.

**Reasons To Reject:**

1)The contribution of the paper is incremental. Some techniques mentioned in the paper, E.g., GRN, Asymmetric Loss, are adapted from other research directions.

2) The specific relation between the design of the model and the Non-Convergent Discourse Parsing task needs to be clarified further.

3) It would be better if a case study could be included in the experimental part.

**Reproducibility:**

3: Could reproduce the results with some difficulty. The settings of parameters are underspecified or subjectively determined; the training/evaluation data are not widely available.

**Reviewer Confidence:**

3: Pretty sure, but there's a chance I missed something. Although I have a good feel for this area in general, I did not carefully check the paper's details, e.g., the math, experimental design, or novelty.

---

> ### Author Rebuttal · Authors · 2023-08-28
>
> *General*: We thank the reviewers for their time and comments and we are happy that all reviewers agree about the soundness, reproducibility and other merits of the work.
>
> Detailed response:
>
> *Incremental*: Allowing ourselves to be argumentative - we actually believe that the contribution is more that incremental, in spite of the integration of borrowed techniques. We believe that the combination of the techniques is unique, and that the overall results pave the way to further research.
>
> *Model design and task*: This is a very good point! On the one hand - we did designed the model with a complex discourse parsing task in mind. On the other hand, we believe that the relation between the architecture and specific task, especially compared to other dialogue-related tasks actually warrants a full paper (that we will be happy to write or read).
>
> *Case-study*: Indeed! Unfortunately we reached the page limit. We plan to add this in the camera ready version if space permits.

---

### Official Review · Reviewer_Ks1Q · 2023-08-17

**Soundness:** 3

**Excitement:**

3: Ambivalent: It has merits (e.g., it reports state-of-the-art results, the idea is nice), but there are key weaknesses (e.g., it describes incremental work), and it can significantly benefit from another round of revision. However, I won't object to accepting it if my co-reviewers champion it.

**Missing References:**

I'm not aware of the literature in this area.

**Paper Topic And Main Contributions:**

The paper introduces an architectural methodology for contentious dialogue parsing. The proposed method is based on roberta, passing the dialogue context and previous labels through GRN layers and use an asymmetric loss to train their model. They compare their approach with one baseline (which uses naive methods, side information and oracle labels) and show improved / similar performance.

**Questions For The Authors:**

* Is the cross entropy in table 1 weighted?
* I think i missed but i couldn't find any ablation results for the auxiliary task. In page 8 first half last para you talk about unimpressive contribution - then why not just remove it and keep things simple?

**Reasons To Accept:**

* The paper is generally well written and easy to follow. The problem is motivated well.
* The approach is well justified and results look sound and reproducible.

**Reasons To Reject:**

* The approach seems primitive in that they only use a single roberta model. I'm wondering how encoder decoder models (T5, Bart, etc) pretrained on dialogue data perform in this task.
* There are works doing discourse parsing (for persuasion and negotiation) which aim to label each utterance in a dialogue. This work fails to compare with the methods employed in those even though the label set (or the specific problem) is different. (https://arxiv.org/abs/1906.06725, https://arxiv.org/abs/2106.00920, etc -- I'm not aware of the latest literature to be honest but these are some old papers I could find).
* What if you directly pass previous utterances to the encoder? Are there any ablations to show the need of the GRN?
* Results lack statistical significance tests.

**Reproducibility:**

5: Could easily reproduce the results.

**Reviewer Confidence:**

2: Willing to defend my evaluation, but it is fairly likely that I missed some details, didn't understand some central points, or can't be sure about the novelty of the work.

**Typos Grammar Style And Presentation Improvements:**

Please use the standard acl submitting style guide which has line numbers in the future.

---

> ### Author Rebuttal · Authors · 2023-08-28
>
> *General*: We thank the reviewers for their time and comments and we are happy that all reviewers agree about the soundness, reproducibility and other merits of the work.
>
> Detailed response:
>
> *Approach, Encoder-Decoders*: This work serves as an initial step demonstrating the applicability of a single architecture, comparing to the cumbersomeness of the approach used in  previous work. The main motivation is not only to achieve better results but to allow a comparison. Indeed, future work should utilize other seq2seq models.
>
> *Prior work about persuasion*: As the reviewers noted - the tagset is very different. Moreover, our motivation is somewhat broader than just labeling dialogue acts for persuasiveness - on the contrary, we aim to understand the benefits of  “failed” persuasion (=non-convergent discussions).
>
> *Passing previous utterances directly+ablation*: We have these results but had to exclude them due to space constraints. We will be reporting on these given the extra page granted in the camera ready.
>
> *CE*: We reported with unweighted BCE. We also experimented with a weighted version but didn’t observe any significant difference.
>
> *Auxiliary task*: It is a common and straightforward approach that was expected to improve results. We report on its shortcomings because we believe it is a surprising negative result. We also wanted to vent our frustration :)

---

### Meta-Review · Area_Chair_p97g · 2023-09-16

**Recommendation:** 3

**Metareview:**

This work proposes an autoregressive method for Non-convergent discourse parsing. The approach and method are interesting, the paper's clarity is notable, and experimentation is on topic and shows promising results. As noted by the reviewers, the main issue is that the method could be evaluated in a more general setting (not just Non-convergent discourse). Therefore, it might have a more significant impact or be described in a larger context than the current version. However, the current version could be published and be a starting point to study the potential and limitations of the proposed approach.

Pros
* Interesting problem and method
* Promising results
* It is a self-contained manuscript

Cons
* Topic very specific but applicability to a larger scope
* Small dataset

---

### Decision · Program_Chairs · 2023-10-07

**Decision:**

Accept-Main

**Comment:**

This work proposes an autoregressive method for Non-convergent discourse parsing. The approach and method are interesting, the paper's clarity is notable, and experimentation is on topic and shows promising results. As noted by the reviewers, the main issue is that the method could be evaluated in a more general setting (not just Non-convergent discourse). Therefore, it might have a more significant impact or be described in a larger context than the current version. However, the current version could be published and be a starting point to study the potential and limitations of the proposed approach.

Pros
* Interesting problem and method
* Promising results
* It is a self-contained manuscript

Cons
* Topic very specific but applicability to a larger scope
* Small dataset